# Evaluation of Rapid Immunochromatographic Tests for the Direct Detection of Extended Spectrum Beta-Lactamases and Carbapenemases in Enterobacterales Isolated from Positive Blood Cultures

Ahmed S. Keshta,[a,b] Nazik Elamin,[a] ⓘMohammad Rubayet Hasan,[a,c] Andrés Pérez-López,[a,c] Diane Roscoe,[a] Patrick Tang,[a,c] ⓘMohammed Suleiman[a]

[a]Sidra Medicine, Doha, Qatar
[b]School of Medicine, Royal College of Surgeons in Ireland–Bahrain, Busaiteen, Kingdom of Bahrain
[c]Weill Cornell Medical College in Qatar, Doha, Qatar

**ABSTRACT** NG-Test CTX-M MULTI and NG-Test Carba 5 (NG Biotech) are two rapid *in vitro* immunochromatographic assays that are widely used for the detection of the most common extended spectrum beta-lactamases (ESBL) and carbapenemases in Enterobacterales. ESBL and carbapenemases are leading causes of morbidity and mortality worldwide and their rapid detection from positive blood cultures is crucial for early initiation of effective antimicrobial therapy in bloodstream infections (BSI) involving antibiotic-resistant organisms. In this study, we developed a rapid workflow for positive blood cultures for direct identification of Enterobacterales by MALDI-TOF mass-spectrometry, followed by detection of ESBL and carbapenemases using NG-Test CTX-M MULTI and NG-Test Carba 5 (NG Biotech). The workflow was evaluated using Enterobacterales isolates ($n = 114$), primarily *Klebsiella* species ($n = 50$) and *Escherichia coli* ($n = 40$). Compared to the standard testing approach in our institution using BD Phoenix, our new testing approach demonstrates 100% sensitivity and specificity for organism identification and detection of ESBL and carbapenemases. Implementation of a rapid workflow in diagnostic microbiology laboratories will enable more effective antimicrobial management of patients with BSI due to ESBL- and carbapenemase-producing Enterobacterales.

**IMPORTANCE** The incidence of bloodstream infections (BSI) with extended spectrum beta-lactamase (ESBL) producing and carbapenemase producing Enterobacterales (CPE) is increasing at an alarming rate, for which only limited therapeutic options remain available. Rapid identification of these bacteria along with their antibiotic resistance mechanisms in positive blood cultures with Gram-negative bacteria will allow for early initiation of effective therapy and limit the overuse of broad-spectrum antibiotics in BSI (1). In this study we evaluated a combined approach of testing positive blood cultures directly, using MALDI-TOF MS followed by rapid immunochromatographic tests, for the detection of ESBLs and CPEs. Our approach demonstrates 100% sensitivity and specificity for the identification of Enterobacterales and detection of ESBLs and CPEs in positive blood culture with a turnaround time (TAT) of ≤60 min compared to a TAT of 48 h required by conventional culture and susceptibility testing methods.

**KEYWORDS** NG-Test CARBA 5, CTX-M, ESBL, CPO

Extended-spectrum beta-lactamase (ESBL)-producing and carbapenemase-producing Enterobacterales (CPE) have been reported as significant causes of morbidity and mortality worldwide. Over the last decades, the increasing prevalence of ESBL-producing

Address correspondence to Mohammed Suleiman, MSuleiman@sidra.org.

Enterobacterales and CPEs have posed a significant challenge for the management of bloodstream infections (BSI) caused by enteric Gram-negative bacteria (1, 2). The conventional methods for detection, identification and antimicrobial susceptibility testing of microorganisms are labor-intensive, cost-inefficient and time-consuming processes. With the limited antibiotic options available against ESBL-producing Enterobacterales and CPEs, efforts directed toward early identification of these bacteria in BSI are important.

MALDI-TOF MS is an FDA-approved technology that has revolutionized clinical microbiology laboratories and has become the first-line instrument for the identification of organisms. It has been proven as a rapid, sensitive, and cost-effective method in the identification of bacterial, fungal, and mycobacterial pathogens from cultures. In recent years, rapid identification of Enterobacterales directly from positive blood culture broths by MALDI-TOF MS using the Sepsityper kit (Bruker Daltonik, Bremen, Germany) or laboratory-developed methods have been implemented in many clinical laboratories worldwide (3). However, in areas with a high prevalence of ESBL-producing Enterobacterales and CPE, direct organism identification from positive blood culture bottles without rapid detection of antimicrobial resistance may have limited impact on the antimicrobial management of sepsis caused by Gram-negative bacteria (4).

Therefore, in our laboratory, we have developed a rapid workflow for simultaneous identification of bacteria and detection of CTX-M-type ESBL and carbapenemase genes in positive blood cultures with Gram-negative bacteria. Our new approach combines direct identification of pathogens in blood culture by MALDI-TOF using an in-house developed processing method with two newly marketed immunochromatographic assays (ICA) for the detection of CTX-M-type ESBL and carbapenemase genes. The performance of the two ICAs assessed in this study, NG-Test CTX-M MULTI and NG-Test Carba 5 (NG Biotech, Guipry, France), have been validated on bacterial colonies grown on solid media (5–6). NG-Test CTX-M MULTI can detect CTX-M-type enzymes belonging to group 1, which includes CTX-M-15, the most prevalent ESBL in our setting and throughout the world (7). In addition, the assay can also detect less prevalent ESBLs belonging to groups 2, 8, 9, and 25. NG-Test CARBA 5 can detect one or more of the five most common carbapenemase enzymes found in Enterobacterales: KPC (K), OXA-48-like (O), IMP (I), VIM (V), NDM (N). This study is aimed to assess the diagnostic yield of NG-Test CTX-M MULTI and NG-Test Carba 5 performed directly on pediatric positive blood cultures for Enterobacterales, and compare the diagnostic performance of our new rapid diagnostic workflow to that obtained using conventional automated susceptibility methods and molecular techniques.

## RESULTS

We have utilized both simulated and positive original blood culture bottles for evaluation of our workflow that starts with the direct identification of the bacteria by MALDI-TOF, which was previously validated, in-house, against the standard identification methods (data not shown). After direct identification by MALDI-TOF, a total of 65 simulated positive blood culture bottles and 49 unknown positive blood culture bottles with Enterobacterales were assessed by NG-Test CTX-M MULTI and NG-Test Carba 5 (NG Biotech) for rapid detection of ESBLs and CPEs, and the results were compared with the results of standard testing approach based on BD Phoenix 100 (Becton, Dickinson, Franklin Lakes, NJ) and/or molecular assays (see methods). Different sets of spiked BD Bactec bottles were used for the verification of respective ICA kits (Table S1). However, for the unknown positive blood culture bottles, the same bottles were tested by both ICA kits prospectively. The isolates assessed were *Klebsiella* species (*n* = 50), *Escherichia coli* (*n* = 40), *Serratia marcescens* (*n* = 12), *Enterobacter cloacae* (*n* = 8), *Salmonella* species (*n* = 3), and *Proteus mirabilis* (*n* = 1).

As illustrated in Table 1, a total of 81 NG-Test CTX-M Multi and 82 NG-Test Carba 5 tests were performed on simulated or unknown samples (see Table S1 for details). NG-Test Carba 5 kit detected 19 carbapenemase producing isolates in 82 positive blood cultures. As compared to the standard test interpretations by the BD Phoenix system, the NG-Test

**TABLE 1** Performance characteristics of direct NG-Test CTX-M MULTI and NG-Test Carba 5 tests on positive blood cultures with reference to standard tests

| Tests (n)[a] | Sample description[b] | Enzyme/s | Results (no.) | | | | 95% CI of the 100% sensitivity | | 95% CI of the 100% specificity | |
|---|---|---|---|---|---|---|---|---|---|---|
| | | | TP | FP | FN | TN | Low | High | Low | High |
| NG-Test CTX-M MULTI (81) | 32 simulated and 49 unknown positive blood culture bottles | CTX-M | 26 | 0 | 0 | 55 | 86.77 | 100 | 93.51 | 100 |
| NG-Test Carba 5 (82) | 33 simulated and 49 unknown positive blood culture bottles | OXA-48, KPC, NDM, VIM1, IMP | 19 | 0 | 0 | 63 | 82.35 | 100 | 94.31 | 100 |

[a]n = number of tests performed with each kit.
[b]Total number of samples including simulated and unknown samples utilized for evaluation is 114; 32 simulated samples tested only with NG-Test CTX-M MULTI kit, 33 different samples tested only with NG-Test Carba 5 kit, 49 unknown samples were prospectively tested with both kits.

Carba 5 kit correctly detected 5 OXA-48-like, 1 KPC-like, 5 NDM-like, and 1 IMP-like isolates, as well as isolates with multiple carbapenemase genes (2 IMP-like + VIM-like, and 3 OXA-48-like + NDM-like). Overall, the NG-Test Carba 5 kit showed 100% sensitivity (95% confidence interval [CI], 82.35% to 100%) and 100% specificity (95% CI, 94.31% to 100%). Similarly, NG-Test CTX-M MULTI detected all isolates (26 out of 81 positive blood cultures) with ESBL activity with 100% sensitivity (95% CI, 86.77% to 100%) and 100% specificity (95% CI, 93.51% to 100%) compared to BD Phoenix 100 results (Table 1). NG-test results were also in 100% agreement with commercial PCR results for the respective genes whenever PCR data were available (Table 1 and Table S1).

## DISCUSSION

Our new rapid diagnostic testing approach demonstrated to be fast, accurate and user-friendly for the identification of bacteria, and detection of CTX-M-type and carbapenemase genes in positive pediatric blood culture bottles with Enterobacterales. However, the theoretical benefit of rapid organism identification together with rapid detection of antimicrobial resistance markers to decrease length of hospital stay and improve mortality in BSI caused by Enterobacterales is yet to be determined (3). The actual clinical impact of our rapid diagnostic approach lies on improving antimicrobial stewardship (ASP) by shortening the time to optimal antimicrobial therapy in settings with high prevalence of antimicrobial resistance among enteric Gram-negative bacteria (8).

The phenotypic detection of antimicrobial resistance mechanisms through manual and automated susceptibility methods can be labor-intensive and may require up to 48 h to results. Although multiplex PCR methods are widely available, their cost can be as high as six times of the cost of the kits used in our evaluation. The use of one standard preparation to obtain bacterial pellets from pediatric positive blood culture broths to perform three tests (MALDI-TOF, NG-Test CTX-M MULTI and NG-Test Carba 5) simultaneously represents a novel approach that provides accurate and rapid results within 60 min after an automated system flags a blood culture bottle as positive. Additionally, the use of both of these ICA kits provides a more cost-effective solution for clinical microbiology laboratories without compromising the quality of the results. Likewise, the initial rapid identification of pathogen from positive blood cultures bottles by MALDI-TOF in our workflow ensures that rapid ICA are only performed on Enterobacterales providing further cost savings.

Our study has several limitations. For example, the test results from the ICA kits for some isolates could not be verified by the identification of specific resistance markers by PCR. Also, our approach is not capable of detecting all resistance mechanisms in a comprehensive manner. For example, ESBLs other than CTX-M-type or plasmid-mediated AmpC beta-lactamases cannot be detected by our approach. To circumvent this limitation, we do not currently report negative test results for resistant markers until phenotypic data from the standard methods are available. Another limitation of our method is that it is labor-intensive compared to new multiplex molecular methods that can be performed in a few simple steps only. But in our setting, with low number of positive blood cultures (2–3 per day), this approach provided a reagent cost saving

of 75%, in comparison with commercial multiplex RT-PCR panels that can detect resistance markers directly from positive blood cultures without adding much extra workload to our technologists. Laboratories with high workload may need to reevaluate this approach in light of reagent versus labor costs.

In conclusion, the combination of MALDI-TOF MS for organism identification with NG-Test CTX-M MULTI and NG-Test Carba 5 tests directly on positive pediatric blood culture bottles represents an accurate, user friendly and inexpensive approach. Our approach coupled with real-time ASP interventions can potentially improve clinical decision-making in Gram-negative sepsis caused by ESBL-producing Enterobacterales and CPE by decreasing substantially the time to effective therapy.

## MATERIALS AND METHODS

**Blood culture system.** The blood cultures were incubated in the BD Bactec FX blood culture system (Becton, Dickinson, Franklin Lakes, NJ). A total of 114 BD Bactec Peds Plus/F bottles (Becton, Dickinson, Franklin Lakes, NJ) were evaluated according to the standard workflow developed in our laboratory in compliance with the Clinical and Laboratory Standards Institutes (CLSI) guidelines (9). Prior to the implementation of the test in our laboratory, 32 bottles were spiked and tested using NG-Test CTX-M MULTI kit, and another 33 different bottles were spiked and tested using NG-Test Carba 5. For spiking, known clinical isolates from previous BSI were used. After the implementation of the test in our laboratory, an additional 49 positive blood cultures were prospectively evaluated and tested simultaneously by using both ICA kits.

**Preparation of simulated blood culture bottles.** Clinical isolates of Enterobacterales previously recovered from positive blood cultures in our laboratory were subcultured in Blood agar medium and incubated for 24 h at 35℃. Bacterial suspensions were prepared in 5 ml normal saline (0.85% NaCl) with a turbidity equivalent to 1 McFarland standard and further diluted 1,000-fold in normal saline. 50 $\mu$l of the diluted preparation was then injected into a new BD Bactec Peds Plus/F bottle (Becton, Dickinson, Franklin Lakes, NJ) using a 1 ml syringe under sterile conditions. In addition, 3 ml of random human blood collected for transfusion was injected into the blood culture bottle as specimen matrix. The blood culture bottle was then incubated in the BD Bactec system until the bottle is flagged positive by the system.

**Gram stain and subculture.** Immediately after a blood culture is signaled positive by BD Bactec FX blood culture system, a Gram stain was prepared. Also, positive blood cultures were streaked into BAP, MAC, and CHOC plates. If the Gram stain showed Gram negative rods, blood cultures were processed for direct identification by MALDI-TOF and rapid detection of ESBLs and CPEs as illustrated below.

**Blood culture pre-processing, direct MALDI-TOF and NG-test.** Blood culture broths from the positive, BD Bactec bottles were drawn using a 10-ml syringe to prepare three 1.5 ml aliquots for simultaneous processing (Fig. 1). To lyse the blood cells, 0.3 ml of 5% SDS was added to each of the aliquots. The tubes were vortexed, incubated at room temperature for 3–5 min, and then centrifuged for 2 min at 13,000g. The supernatant was removed, and the pellet was resuspended in 1 ml of water. Centrifugation and supernatant removal cycles were continued three more times to wash the bacterial pellet with 1 ml of water, 1 ml of 70% ethanol and 0.3 ml of water, respectively. From one aliquot, 1 $\mu$l of the bacterial pellet was used to inoculate the MALDI-TOF target plate and was allowed to dry. Next, 1 $\mu$l of 70% formic acid was added and allowed to dry. Finally, 1 $\mu$l of HCCA matrix ($\alpha$-Cyano-4-hydroxycinnamic acid) was added and allowed to dry before performing MALDI-TOF (Bruker Daltonik, Bremen, Germany) testing according to the manufacturer's instructions. If a member of the Enterobacterales was identified by MALDI-TOF, the NG-test CTX-M MULTI and NG-test CARBA 5 were performed on the pellets from the second and third aliquots. Five drops (approximately 150 $\mu$l) of the extraction buffer from the kit was added to the entire volume of the pellet in 1.5 ml microcentrifuge tubes and vortexed to homogenize the mixture. Finally, using the pipette provided with the kit, 100 $\mu$l of the prepared mixture was added to the testing cassette and the results were read after 15 min. On the following day, the NG-test CTX-M MULTI and NG-test CARBA 5 tests were repeated on isolated colonies according to manufacturer's instructions.

**Result comparison against reference methods.** All procedures for standard testing were performed according to CLSI guidelines (9). The results were compared with the results from standard testing by BD Phoenix 100 (Becton, Dickinson, Franklin Lakes, NJ) on isolated colonies. Some isolates were randomly tested using one/or two multiplex real-time PCR assays: Eazyplex CRE Superbug (Amplex BioSystems, Giessen, Germany) for ESBL producers and Xpert Carba-R assay (Cepheid, Sunnyvale, CA, USA) for carbapenemase producers (Table S1 and S2) from isolated colonies. Antibiotic susceptibility testing was performed on the BD Phoenix 100 instrument using the Gram negative panel (NMIC-501) panel, ESBL test, CPO Detect test on isolated colonies according to the manufacturer's instructions. CPO Detect test is a qualitative confirmatory growth-based test embedded in Gram-negative NMIC-502 panels for detection and confirmation of class A, B and D carbapenemases, which was used in this study to confirm NG Test Carba 5 results (10). The Phoenix ESBL test uses the BDXpert rules integrated with growth response to selected extended-spectrum (cefpodoxime) and broad-spectrum (ceftazidime, ceftriaxone, cefotaxime) cephalosporins, with or without clavulanic acid, to detect the production of ESBL, which was used in this study to confirm NG Text CT-M Multi results (11). PCR confirmation testing for CTX-M was performed on the Amplex instrument (AmplexDiagnostics GmbH) using the Easyplex SuperBug CRE kit on isolated colonies. PCR testing confirmation for carbapenemases was performed on the GeneXpert system using the Xpert Carba-R kit (Cepheid) on isolated colonies per manufacturer's instructions.

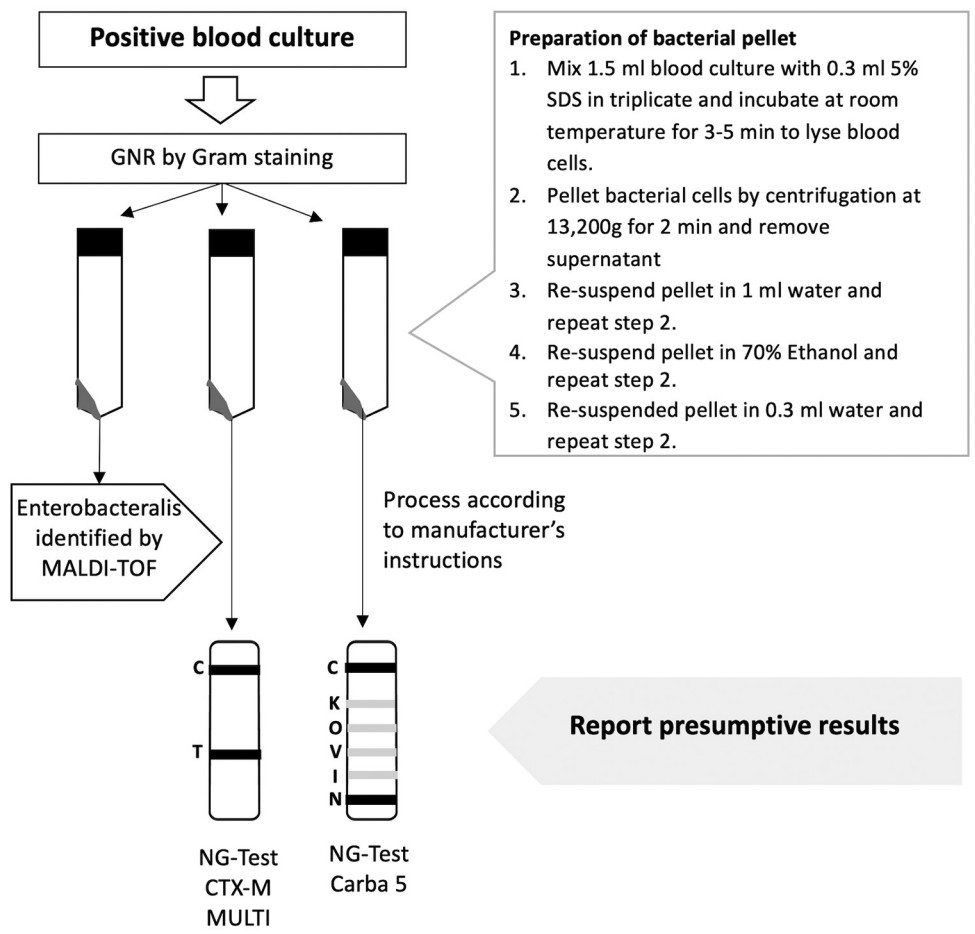

**FIG 1** Workflow for NG-Test CTX-M MULTI and NG-Test Carba 5 tests on positive blood cultures in combination with MALDI-TOF MS. GNR, Gram negative rods; MALDI-TOF, matrix-assisted laser desorption ionization-time of flight; SDS, sodium dodecyl sulfate.

## SUPPLEMENTAL MATERIAL

Supplemental material is available online only.
**SUPPLEMENTAL FILE 1**, PDF file, 0.3 MB.

## ACKNOWLEDGMENTS

We would like to express our special thanks to the microbiology laboratory technologists at Sidra Medicine who performed this testing to provide quality and timely results to our patients.

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
