## [Reviewer comments · Microbiology Spectrum]

Microbiology Spectrum

Evaluation of rapid immunochromatographic tests for the direct detection of extended spectrum beta-lactamases and carbapenemases in Enterobacterales isolated from positive blood cultures

Ahmed Keshta, Nazik Elamin, Mohammad Hasan, Andres Perez-Lopez, Diane Roscoe, Patrick Tang, and Mohammed Suleiman

Corresponding Author(s): Mohammed Suleiman, Sidra Medical & Research Center

Review Timeline:

Submission Date:	July 10, 2021
Editorial Decision:	July 19, 2021
Revision Received:	August 9, 2021
Editorial Decision:	September 14, 2021
Revision Received:	November 1, 2021
Editorial Decision:	November 4, 2021
Revision Received:	November 7, 2021
Accepted:	November 10, 2021

Editor: William Lainhart

Reviewer(s): Disclosure of reviewer identity is with reference to reviewer comments included in decision letter(s). The following individuals involved in review of your submission have agreed to reveal their identity: Janet A Hindler (Reviewer #1)

Transaction Report:

DOI: <https://doi.org/10.1128/Spectrum.00785-21>

July 19, 2021

Mr. Mohammed Suleiman
Sidra Medical & Research Center
Doha
Qatar

Re: Spectrum00785-21 (Evaluation of rapid immunochromatographic tests for the direct detection of extended spectrum beta-lactamases and carbapenemases in Enterobacterales from positive blood cultures)

Dear Mr. Mohammed Suleiman:

Thank you for submitting your manuscript to Microbiology Spectrum. When submitting the revised version of your paper, please provide (1) point-by-point responses to the issues raised by the reviewers as file type "Response to Reviewers," not in your cover letter, and (2) a PDF file that indicates the changes from the original submission (by highlighting or underlining the changes) as file type "Marked Up Manuscript - For Review Only". Please use this link to submit your revised manuscript - we strongly recommend that you submit your paper within the next 60 days or reach out to me. Detailed information on submitting your revised paper are below.

Link Not Available

Sincerely,

William Lainhart

Journals Department
Editor comments:

After conferring with senior editors and editorial staff, we suggest that this article be reformatted and resubmitted as either a research article or as a Methods and Protocols article (<https://journals.asm.org/journal/spectrum/article-types>). At that point, after reformatting and resubmission, the article will be evaluated and sent for review.

Staff Comments:

Preparing Revision Guidelines

- Point-by-point responses to the issues raised by the reviewers in a file named "Response to Reviewers," NOT IN YOUR COVER LETTER.
- Upload a compare copy of the manuscript (without figures) as a "Marked-Up Manuscript" file.
- Each figure must be uploaded as a separate file, and any multipanel figures must be assembled into one file.

- Manuscript: A .DOC version of the revised manuscript
- Figures: Editable, high-resolution, individual figure files are required at revision, TIFF or EPS files are preferred

For complete guidelines on revision requirements, please see the Instructions to Authors at [link to page]. **Submissions of a paper that does not conform to Microbiology Spectrum guidelines will delay acceptance of your manuscript.**

Please return the manuscript within 60 days; if you cannot complete the modification within this time period, please contact me. If you do not wish to modify the manuscript and prefer to submit it to another journal, please notify me of your decision immediately so that the manuscript may be formally withdrawn from consideration by Microbiology Spectrum.

If you would like to submit an image for consideration as the Featured Image for an issue, please contact Spectrum staff.

September 14, 2021

Mr. Mohammed Suleiman
Sidra Medical & Research Center
Doha
Qatar

Re: Spectrum00785-21R1 (Evaluation of rapid immunochromatographic tests for the direct detection of extended spectrum beta-lactamases and carbapenemases in Enterobacterales isolated from positive blood cultures)

Dear Mr. Mohammed Suleiman:

Thank you for submitting your manuscript to Microbiology Spectrum. When submitting the revised version of your paper, please provide (1) point-by-point responses to the issues raised by the reviewers as file type "Response to Reviewers," not in your cover letter, and (2) a PDF file that indicates the changes from the original submission (by highlighting or underlining the changes) as file type "Marked Up Manuscript - For Review Only". Please use this link to submit your revised manuscript - we strongly recommend that you submit your paper within the next 60 days or reach out to me. Detailed information on submitting your revised paper are below.

Link Not Available

Sincerely,

William Lainhart

Journals Department
Reviewer comments:

Reviewer #1 (Comments for the Author):

The authors describe a workflow approach for direct testing of bacterial growth from positive blood cultures with MALDI-TOF and two rapid immunochromatographic assays for detection of ESBLs (ie CTX-M) and carbapenemases (KPC, NDM, IMP, VIM, OXA-48). A total of 163 blood culture specimens demonstrating gram-negative rods were evaluated. Results from the immunochromatographic assays performed on growth confirmed as Enterobacterales were compared to those obtained from BD Phoenix panels and commercial PCR kits for ESBLs and carbapenemases.

Major Comments:

1. Preparation of inocula from positive blood cultures described here as an alternative workflow approach to obtain more rapid results from MALDI-TOF and the two immunochromatographic assays appears sound. However, there is only speculative or theoretical information as to the value of this approach to patient care in the authors setting. Perhaps the emphasis should be on the methods, but the methods need clarification (please see below). Also, although the immunochromatographic methods may be less costly than some molecular methods, some would not find the labor involved in this approach workable in their setting. Therefore, it would be helpful to include some pros and cons compared to other direct detection methods.
2. Although many laboratories use direct detection methods for positive blood cultures and have demonstrated improved outcomes, the value of resistance marker testing has its limitations, especially for negative test results. If authors elect to retain

the speculative or theoretical value of the method they describe, these should be discussed.

Minor Comments

3. Line 46 - please provide a reference that describes the impact on therapy of direct detection of resistance mechanisms in positive blood cultures
4. Line 49 - it would be helpful to indicate actual time savings versus "much faster".
5. Line 77 - authors may wish to consider rephrasing this comment since there are advantages to use of conventional or phenotypic methods (e.g., can test many drug classes that might be considered for BSI) and some disadvantages of tests that only detect limited ESBLs and carbapenemases.
6. Line 86 - please add reference for "throughout the world".
7. Line 95 - the abstract indicates the workflow was evaluated using 163 Enterobacteriales so unclear why only *E. coli* and *Klebsiella pneumoniae* (n=73) are mentioned separately in this first sentence in the Results? Please see below suggestions for expanding Methods section.
8. Line 102 - please clarify what is meant by "retrospective and prospective" assessment (Methods section?).
9. Line 117 - it would be helpful to explain why PCR data might not be available for all isolates (Methods section?)
10. Line 128 - please reference what would be considered high prevalence
11. Line 130 - please quantify "a significant proportion". Also, perhaps clarify the following statement that ceftriaxone / cefotaxime may not be an empiric choice if a "significant proportion" of GNR are ESBL producers.
12. Line 136 - authors might mention other mechanisms of resistance to expanded-spectrum cephalosporins that must be factored into de-escalation decisions if CTX-M is not detected. Also, it may be helpful to indicate other factors in addition to expenditures that must be considered regarding overuse of carbapenems. And authors might mention phenotypic results (e.g. MICs) versus presence or absence of a resistance mechanism are an important consideration when tailoring therapy.
13. Line 165 - authors provide no data on costs so it is difficult to appreciate their comment about this being an "inexpensive approach".
14. Methods - please clarify the numbers of tests/isolates throughout and methods used. Include numbers of isolates and methods used for validation of MALDI (as described in line 99) and the two immunochromatographic methods. Clarify, if true, that the 163 positive blood cultures were from single patients and also that these were not spiked cultures. Even though there was 100% sensitivity and 100% specificity, it is important to include all details of the testing or provide references to support procedures used.
15. Line 173 - please clarify "retrospectively" and "prospectively"
16. Line 175 - please provide CLSI reference
17. Line 194 - what volume of "pellet" was applied to the assays?
18. Line 197 - please provide CLSI reference
19. Line 197 - were isolated colonies used for BD Phoenix and PCR assays? Were only pellets used for the immunochromatographic assays (ie, test was only done on pellets and not repeated on isolated colonies as recommended by the manufacturer)?
20. Line 203 - it would be helpful to add a sentence or two explaining the CPO Detect test since many may not know about this unique test on the Phoenix panel. It is also important to explain the "reference results" from BD Phoenix that were used for the comparisons for both CTX-M and NG-Test Carba 5 assays.
21. Line 206 - please confirm that testing with all kits was performed following the manufacturers' instructions, if true
22. Table 1 - please clarify the total number of positive blood cultures tested (163?) and clarify the n=81 and n=82. If not all samples were tested with both assays, how was it decided which would be tested with CTX-M versus Carba 5? Also, it would be helpful to breakdown this table by organism species and separate out the carbapenemase genes and where 2 genes were present. It is apparent this information is in the supplemental tables but it would be most helpful to have it in the body of the text, if possible.
23. Supplemental Tables - again, please clarify numbering. Here one might think that isolate #1 *E. coli* has CTX-M (Table 1) and OXA-48 (Table 2). Perhaps use different numbers for specimens 1-163.
24. Supplemental Tables - it would be helpful to clarify (footnote would be acceptable) for inoculum source for all 3 tests (e.g., pellet, isolated colonies)
25. Supplemental Tables - were MALDI IDs of *Klebsiella* only to species level? (e.g. *Klebsiella* spp.). The text references *Klebsiella pneumoniae* in several places??
26. Supplemental Table 1 - it is unclear why EasyPlex was not done on all samples, especially *Klebsiella* spp. isolates #31 and #32 and *E. cloacae* #62. For these, how was it determined that CTX-M was "confirmed"?

Staff Comments:

Preparing Revision Guidelines

To submit your modified manuscript, log onto the eJP submission site at <https://spectrum.msubmit.net/cgi-bin/main.plex>. Go to Author Tasks and click the appropriate manuscript title to begin the revision process. The information that you entered when you first submitted the paper will be displayed. Please update the information as necessary. Here are a few examples of required

updates that authors must address:

Please return the manuscript within 60 days; if you cannot complete the modification within this time period, please contact me. If you do not wish to modify the manuscript and prefer to submit it to another journal, please notify me of your decision immediately so that the manuscript may be formally withdrawn from consideration by Microbiology Spectrum.

Re: Spectrum00785-21R1 (Evaluation of rapid immunochromatographic tests for the direct detection of extended spectrum beta-lactamases and carbapenemases in Enterobacterales isolated from positive blood cultures)

We would like to thank the reviewers for taking the time to review this work and to provide constructive suggestions to improve the quality of our manuscript. Below, please find our point by point responses to the reviewers' comments:

Reviewer's major comments	Author's response
1- Preparation of inocula from positive blood cultures described here as an alternative workflow approach to obtain more rapid results from MALDI-TOF and the two immunochromatographic assays appears sound. However, there is only speculative or theoretical information as to the value of this approach to patient care in the authors setting. Perhaps the emphasis should be on the methods, but the methods need clarification (please see below). Also, although the immunochromatographic methods may be less costly than some molecular methods, some would not find the labor involved in this approach workable in their setting. Therefore, it would be helpful to include some pros and cons compared to other direct detection methods.	We agree with the reviewer that the main focus of our study is the evaluation of a workflow that utilizes MALDI-TOF and rapid immunochromatographic tests against standard identification and antibiotic susceptibility tests. Therefore, we have removed the speculative discussion on potential clinical impact from the discussion section in the revised manuscript (Lines 129 to 147 from previous submitted manuscript removed) Furthermore, we have elaborated the method section as per suggestions from the reviewer.
2- Although many laboratories use direct detection methods for positive blood cultures and have demonstrated improved outcomes, the value of resistance marker testing has its limitations, especially for negative test results. If authors elect to retain the speculative or theoretical value of the method they describe, these should be discussed.	This issue has been included in the revised manuscript as a limitation of the study (lines 143 to 155).

Reviewer's minor comments	Author's response
3- Line 46 - please provide a reference that describes the impact on therapy of direct detection of resistance mechanisms in positive blood cultures	Reference added
4- Line 49 - it would be helpful to indicate actual time savings versus "much faster".	We have added a brief comparison of expected time savings using our method and conventional culture and susceptibility methods (lines 47-50).

5- Line 77 - authors may wish to consider rephrasing this comment since there are advantages to use of conventional or phenotypic methods (e.g., can test many drug classes that might be considered for BSI) and some disadvantages of tests that only detect limited ESBLs and carbapenemases.	We agree with the reviewer. However, here, by conventional methods, we in fact meant direct MALDI-TOF identification alone, not the standard identification and susceptibility methods. For better clarity we have now rephrased this sentence in the revised manuscript (lines 75-77)
6- Line 86 - please add reference for "throughout the world".	Reference added
7- Line 95 - the abstract indicates the workflow was evaluated using 163 Enterobacteriales so unclear why only E. coli and Klebsiella pneumoniae (n=73) are mentioned separately in this first sentence in the Results? Please see below suggestions for expanding Methods section.	We apologize for the confusion. In the revised manuscript, first we have corrected the total number of Enterobacteriales isolates after removing redundant isolates in the abstract. Next, we have modified the respective sections in the results section to accurately reflect the number of isolates under different categories (lines 102-106)
8- Line 102 - please clarify what is meant by "retrospective and prospective" assessment (Methods section?).	We apologize for the confusion. A total of 65 simulated samples (in other words blood culture bottles spiked with previously cultured Enterobacteriales isolates from BSI) and a total of 49 prospectively collected unknown positive blood cultures were assessed in this study. We have clarified these in the revised manuscript (lines 94-98).
9- Line 117 - it would be helpful to explain why PCR data might not be available for all isolates (Methods section?)	It is true that PCR data is not available for all samples assessed in this study. However, the gold standard, reference method in our study is BD Phoenix 100, with which we compared the results from the rapid immunochromatographic tests. PCR data available for some isolates were provided as supporting evidence. We have mentioned this limitation in the revised manuscript (lines 143-144).
10- Line 128 - please reference what would be considered high prevalence	Reference added
11- Line 130 - please quantify "a significant proportion". Also, perhaps clarify the following statement that ceftriaxone / cefotaxime may not be an empiric choice if a "significant proportion" of GNR are ESBL producers.	We have removed these sentences in the revised manuscript in order to address the first major comment from this reviewer, which appeared speculative and beyond the scope of the present study.
12- Line 136 - authors might mention other mechanisms of resistance to expanded-spectrum cephalosporins that must be factored into de-escalation decisions if CTX-M is not detected. Also, it may be helpful to indicate other factors in addition to expenditures that must be considered regarding overuse of carbapenems. And authors might mention phenotypic results (e.g. MICs) versus presence or absence of a resistance mechanism are an important consideration when tailoring therapy.	We agree with the reviewer and added these points as the limitation of our study.(lines 143 to 155).

13- Line 165 - authors provide no data on costs so it is difficult to appreciate their comment about this being an "inexpensive approach".	We have added a statement comparing the cost of our approach against the cost of commercial multiplex RT-PCR panels (Lines 133 and 152)
14- Methods - please clarify the numbers of tests/isolates throughout and methods used. Include numbers of isolates and methods used for validation of MALDI (as described in line 99) and the two immunochromatographic methods. Clarify, if true, that the 163 positive blood cultures were from single patients and also that these were not spiked cultures. Even though there was 100% sensitivity and 100% specificity, it is important to include all details of the testing or provide references to support procedures used.	We have modified and elaborated the methods section to clarify these points (Lines 165 to 173)
15- Line 173 - please clarify "retrospectively" and "prospectively"	We apologize for the confusion. We have rephrased the statement to avoid confusion (Lines 165 to 173) Also please see our response to minor comment 8 above.
16- Line 175 - please provide CLSI reference	Reference added
17- Line 194 - what volume of "pellet" was applied to the assays?	Entire volume of the pellet was used. We have added this information in the revised manuscript (Lines 200 to 205)
18- Line 197 - please provide CLSI reference	Reference added
19- Line 197 - were isolated colonies used for BD Phoenix and PCR assays? Were only pellets used for the immunochromatographic assays (ie, test was only done on pellets and not repeated on isolated colonies as recommended by the manufacturer)?	Confirmatory testing using BD Phoenix and PCR methods was performed on isolated colonies. The immunochromatographic tests were repeated from colonies using the kits on the following day. (Clarified in lines 206 to 224)
20- Line 203 - it would be helpful to add a sentence or two explaining the CPO Detect test since many may not know about this unique test on the Phoenix panel. It is also important to explain the "reference results" from BD Phoenix that were used for the comparisons for both CTX-M and NG-Test Carba 5 assays.	A sentence added to define the CPO Detection test with a reference and explained in more details about using Phoenix as reference results (Lines 214 to 221)
21- Line 206 - please confirm that testing with all kits was performed following the manufacturers' instructions, if true	We confirm that all tests were performed according to manufacturer's instructions unless otherwise stated.
22- Table 1 - please clarify the total number of positive blood cultures tested (163?) and clarify the n=81 and n=82. If not all samples were tested with both assays, how was it decided which would be tested with CTX-M versus Carba 5? Also, it would be helpful to breakdown this table by organism species and separate out the carbapenemase genes and where 2 genes were	We have revised this table according to the suggestions provided by the reviewer.

present. It is apparent this information is in the supplemental tables but it would be most helpful to have it in the body of the text, if possible.	
23- Supplemental Tables - again, please clarify numbering. Here one might think that isolate #1 E. coli has CTX-M (Table 1) and OXA-48 (Table 2). Perhaps use different numbers for specimens 1-163.	New supplementary table created with uniform numbering system to address this comment
24- Supplemental Tables - it would be helpful to clarify (footnote would be acceptable) for inoculum source for all 3 tests (e.g., pellet, isolated colonies)	Footnote added to the new supplementary table
25- Supplemental Tables - were MALDI IDs of Klebsiella only to species level? (e.g. Klebsiella spp.). The text references Klebsiella pneumoniae in several places??	In the new supplementary table, full identification is added
26- Supplemental Table 1 - it is unclear why EasyPlex was not done on all samples, especially Klebsiella spp. isolates #31 and #32 and E. cloacae #62. For these, how was it determined that CTX-M was "confirmed"?	It is true Easyplex data were not available for all isolates . For the 3 isolates in question, we could not confirm the presence of CTX-M gene but we confirmed that they are ESBL producers by manual susceptibility tests (E-tests by Biomerieux). These isolates were flagged as "possible ESBLs" by the BDxpert system. Also, please note our response to minor comment 9.

November 4, 2021

Dr. Mohammed Suleiman
Sidra Medical & Research Center
Doha
Qatar

Re: Spectrum00785-21R2 (Evaluation of rapid immunochromatographic tests for the direct detection of extended spectrum beta-lactamases and carbapenemases in Enterobacterales isolated from positive blood cultures)

Dear Dr. Mohammed Suleiman:

Thank you for submitting your manuscript to Microbiology Spectrum. As you will see your paper is very close to acceptance. Please modify the manuscript along the lines I have recommended. As these revisions are quite minor, I expect that you should be able to turn in the revised paper in less than 30 days, if not sooner. If your manuscript was reviewed, you will find the reviewers' comments below.

When submitting the revised version of your paper, please provide (1) point-by-point responses to the issues I raised in your cover letter, and (2) a PDF file that indicates the changes from the original submission (by highlighting or underlining the changes) as file type "Marked Up Manuscript - For Review Only". Please use this link to submit your revised manuscript. Detailed information on submitting your revised paper are below.

Link Not Available

Sincerely,

William Lainhart

Editor comments:

Please address the following comments and resubmit. Thank you for your hard work on the revision.

Line numbers correspond to those in the "marked-up" document.

Line 107: italicize *Proteus mirabilis*

Line 110: wording of this sentence needs to be addressed for clarity. I am reading it two ways: The Carba 5 detected 19 carbapenemase producing isolates in 82 positive blood cultures? Or, of the 82 cultures positive with carbapenemase producing organisms, Carba 5 only detected 19? Which is correct?

Lines 211-214: How were these isolates chosen for PCR testing?

Table 1: Number of specimens listed under "Sample description" does not match the number listed in column "Tests (n)" and in the TP and TN columns for the NG-Test.

Preparing Revision Guidelines

To submit your modified manuscript, log onto the eJP submission site at <https://spectrum.msubmit.net/cgi-bin/main.plex>. Go to Author Tasks and click the appropriate manuscript title to begin the revision process. The information that you entered when you first submitted the paper will be displayed. Please update the information as necessary. Here are a few examples of required

updates that authors must address:

- point-by-point responses to the issues I raised in your cover letter
- Upload a compare copy of the manuscript (without figures) as a "Marked-Up Manuscript" file.
- Each figure must be uploaded as a separate file, and any multipanel figures must be assembled into one file.
- Manuscript: A .DOC version of the revised manuscript
- Figures: Editable, high-resolution, individual figure files are required at revision, TIFF or EPS files are preferred

Please return the manuscript within 60 days; if you cannot complete the modification within this time period, please contact me. If you do not wish to modify the manuscript and prefer to submit it to another journal, please notify me of your decision immediately so that the manuscript may be formally withdrawn from consideration by Microbiology Spectrum.

Re: Spectrum00785-21R1 (Evaluation of rapid immunochromatographic tests for the direct detection of extended spectrum beta-lactamases and carbapenemases in Enterobacterales isolated from positive blood cultures)

We would like to thank the editor for taking the time to review this work and to provide constructive suggestions to improve the quality of our manuscript. Below, please find our point by point responses to the editor's comments:

Editor's comments	Author's response
1- Line 107: italicize Proteus mirabilis	Thanks, we have corrected it in the revised manuscript. (Line 107)
2- Line 110: wording of this sentence needs to be addressed for clarity. I am reading it two ways: The Carba 5 detected 19 carbapenemase producing isolates in 82 positive blood cultures? Or, of the 82 cultures positive with carbapenemase producing organisms, Carba 5 only detected 19? Which is correct?	We meant that Carba 5 kit detected 19 carbapenemase producing isolates in 82 positive blood cultures. This has been rephrased in the revised manuscript as the reviewer suggested. (Line 109)
3- Lines 211-214: How were these isolates chosen for PCR testing?	The isolates were randomly chosen for confirmation by PCR testing (Line 210)
4- Table 1: Number of specimens listed under "Sample description" does not match the number listed in column "Tests (n)" and in the TP and TN columns for the NG-Test.	We were unable to find any mismatch with the number of tests. However, we have modified the footnotes of Table 1 for further clarification:  • 81 tests performed on CTX-M:  ○ 32 simulated + 49 unknown = 81 ○ 26 TP + 55 TN + 0 FP + 0 FN = 81 • 82 tests performed on Carba 5:  ○ 33 simulated + 49 unknown = 82 ○ 19 TP + 63 TN = 82

Sincerely,

Mohammed Suleiman, MPH, CPHQ, SM, MLS (ASCP)
Department of Pathology, Sidra Medicine
Pathology Clinical Manager – Microbiology, Virology and MID
Level 2M, Office H2M-24093
PO BOX 26999, Doha, Qatar
Phone: +974 40032990; +974 66547242
Email: MSuleiman@sidra.org

November 10, 2021

Dr. Mohammed Suleiman
Sidra Medical & Research Center
Doha
Qatar

Re: Spectrum00785-21R3 (Evaluation of rapid immunochromatographic tests for the direct detection of extended spectrum beta-lactamases and carbapenemases in Enterobacterales isolated from positive blood cultures)

Dear Dr. Mohammed Suleiman:

Your manuscript has been accepted, and I am forwarding it to the ASM Journals Department for publication. You will be notified when your proofs are ready to be viewed.

Sincerely,

William Lainhart
Editor, Microbiology Spectrum
